# Distinct spatial coordinate of visual and vestibular heading signals in macaque FEFsem and MSTd

**Lihua Yang[1,2], Yong Gu[1]\***

[1]Key Laboratory of Primate Neurobiology, Institute of Neuroscience, CAS Center for Excellence in Brain Science and Intelligence Technology, Chinese Academy of Sciences, Shanghai, China; [2]University of Chinese Academy of Sciences, Beijing, China

**Abstract** Precise heading estimate requires integration of visual optic flow and vestibular inertial motion originating from distinct spatial coordinates (eye- and head-centered, respectively). To explore whether the two heading signals may share a common reference frame along the hierarchy of cortical stages, we explored two multisensory areas in macaques: the smooth pursuit area of the frontal eye field (FEFsem) closer to the motor side, and the dorsal portion of medial superior temporal area (MSTd) closer to the sensory side. In both areas, vestibular signals are head-centered, whereas visual signals are mainly eye-centered. However, visual signals in FEFsem are more shifted towards the head coordinate compared to MSTd. These results are robust being largely independent on: (1) smooth pursuit eye movement, (2) motion parallax cue, and (3) behavioral context for active heading estimation, indicating that the visual and vestibular heading signals may be represented in distinct spatial coordinate in sensory cortices.
DOI: https://doi.org/10.7554/eLife.29809.001

## Introduction

To navigate effectively through the environment, we usually combine multiple sensory inputs to precisely estimate our direction of self-motion (i.e, heading). Two most powerful cues are the visual optic flow (*Gibson, 1950*; *Warren, 2003*) and the vestibular inertial motion signals. Numerous psychophysical studies have shown that human and nonhuman primates can improve heading perception by combining optic flow and vestibular cues in a statistically optimal or near optimal way (*Telford et al., 1995*; *Ohmi, 1996*; *Harris et al., 2000*; *Bertin and Berthoz, 2004*; *Gu et al., 2008*; *Butler et al., 2010*, *2011*; *Fetsch et al., 2011*; *Butler et al., 2015*). However, a critical problem for the brain to combine cues is that the two heading signals originate from different spatial reference frames: visual signals arise from the retina and are represented in an eye-centered coordinate, whereas vestibular signals arise from the peripheral organs in the inner ears that are represented in a head-centered coordinate. In this case, the vision information about heading is often confounded by changes of the eye ball in orbits, for example, when subjects vary gaze eccentrically or pursue moving objects (*Royden et al., 1992*; *Royden, 1994*; *Royden et al., 1994*; *Warren and Saunders, 1995*; *Banks et al., 1996*; *Royden and Hildreth, 1996*; *Crowell et al., 1998*). How the brain exactly compensates these eye movements to recover true heading and correctly integrate it with the vestibular cues is unclear.

An intuitive solution to effectively combine visual and vestibular heading cues is to transform them into a common reference frame (*Stein et al., 1993*; *Cohen and Andersen, 2002*). Nevertheless, so far this hypothesis has not been supported by neurophysiological findings in mid-stage sensory areas including the ventral intraparietal area (VIP) and the dorsal portion of the medial superior

**\*For correspondence:**
guyong@ion.ac.cn

**Competing interests:** The authors declare that no competing interests exist.

temporal area (MSTd) (*Avillac et al., 2005*; *Fetsch et al., 2007*; *Takahashi et al., 2007*; *Chen et al., 2013*). Specifically, the spatial reference frame remains largely separated in these areas such that the visual optic flow signals are mainly eye-centered and the vestibular signals are mainly head/body centered. One possibility is that the visual information might be further transformed to be more head-centered when propagated to higher-stage areas. Therefore, in the current study we targets on an area at a later stage in the dorsal visual pathway: the smooth pursuit area of the frontal eye field, that is FEFsem, which receives heavy inputs from MSTd and is closer to the motor side (*Felleman and Van Essen, 1991*; *Fukushima, 2003*; *Lynch and Tian, 2006*). Similar to MSTd, this area also contains robust visual and vestibular signals that may contribute to heading estimation (*Gu et al., 2016*).

Furthermore, there are a number of limitations of methodologies used in previous works that may have led to conflict conclusions from these works. First, some researchers have measured tuning curves in a limited heading range that could have confounded tuning shift versus gain change (*Bradley et al., 1996*; *Bremmer et al., 1997a*; *Bremmer et al., 1997b*; *Page and Duffy, 1999*; *Shenoy et al., 1999*; *Zhang et al., 2004*). Second, people have conducted smooth eye movement compensation experiments (*Bradley et al., 1996*; *Zhang et al., 2004*) that are not necessarily linked with spatial reference frame which should have been measured under different static eye positions (*Fetsch et al., 2007*; *Takahashi et al., 2007*; *Chen et al., 2013*; *Sunkara et al., 2015*). Third, available depth cues such as motion parallax are not consistent across the above studies. Last but not least, people have measured visual tuning curves under passive viewing tasks (*Fetsch et al., 2007*; *Takahashi et al., 2007*; *Chen et al., 2013*; *Sunkara et al., 2015*). In such a behavioral context, the visual signals are not obliged to be transformed towards a head-centered reference frame for heading judgments.

In the current study, we first measured complete visual and vestibular tuning curves from single neurons in FEFsem and MSTd while the animals varied static fixation positions under a passive viewing task. We then reexamined the results under different conditions including: (1) smooth pursuit eye movements, (2) motion parallax within the visual optic flow, and (3) active versus passive behavioral context for heading signals. Our results not only revealed the reference frame properties of visual and vestibular heading signals in FEFsem, but also provided new methodology for studying spatial coordinates of multisensory signals in other brain regions.

## Results

Heading stimuli were delivered through a virtual reality apparatus (*Figure 1A*) that allowed independent control of visual optic flow and vestibular inertial motion cues (see Materials and methods). Visual simulated self-motion, or the physical translation of the body was presented along 8 directions with 45° apart in the horizontal and sagittal planes (*Figure 1B*). A preferred plane was then identified for each isolated neuron for the eccentric fixation task. During the task, monkeys maintained fixation at a central (0°) or eccentric target (20°) while experiencing the heading stimuli. The eccentric targets were either placed in the horizontal meridian for the preferred horizontal plane, or in the vertical meridian for the preferred sagittal plane (*Figure 1C*). During fixation, the animals were head- and body-fixed within the experimental apparatus. Thus, we can distinguish an eye- versus head-centered spatial coordinate in the current study, but cannot distinguish a head- versus body-centered coordinate. For convenience, we simply used head-centered term throughout the text to represent a reference frame that could potentially be head-, body- or even world-centered. We then recorded from well isolated single neurons in the two areas of FEFsem and MSTd (*Figure 1D,E*).

### Spatial coordinate of visual and vestibular signals

We first measured the spatial reference frame of the visual and vestibular signals in FEFsem and MSTd under the eccentric fixation task while the animals passively experienced the heading stimuli. This part of experiment included 171 neurons significantly tuned to visual optic flow and 71 neurons significantly tuned to inertial motion in FEFsem. In MSTd, 99 and 54 neurons that were significantly tuned to optic flow and inertial motion, respectively, were included in the final dataset (*Figure 1E*).

*Figure 2A* shows a typical FEFsem neuron with tuning curves measured under three different eye positions. Qualitatively, this neuron's tuning curve in the visual condition is systematically shifted as a function of the animal's gaze (*Figure 2A*, top panel), while is largely independent on the gaze in the

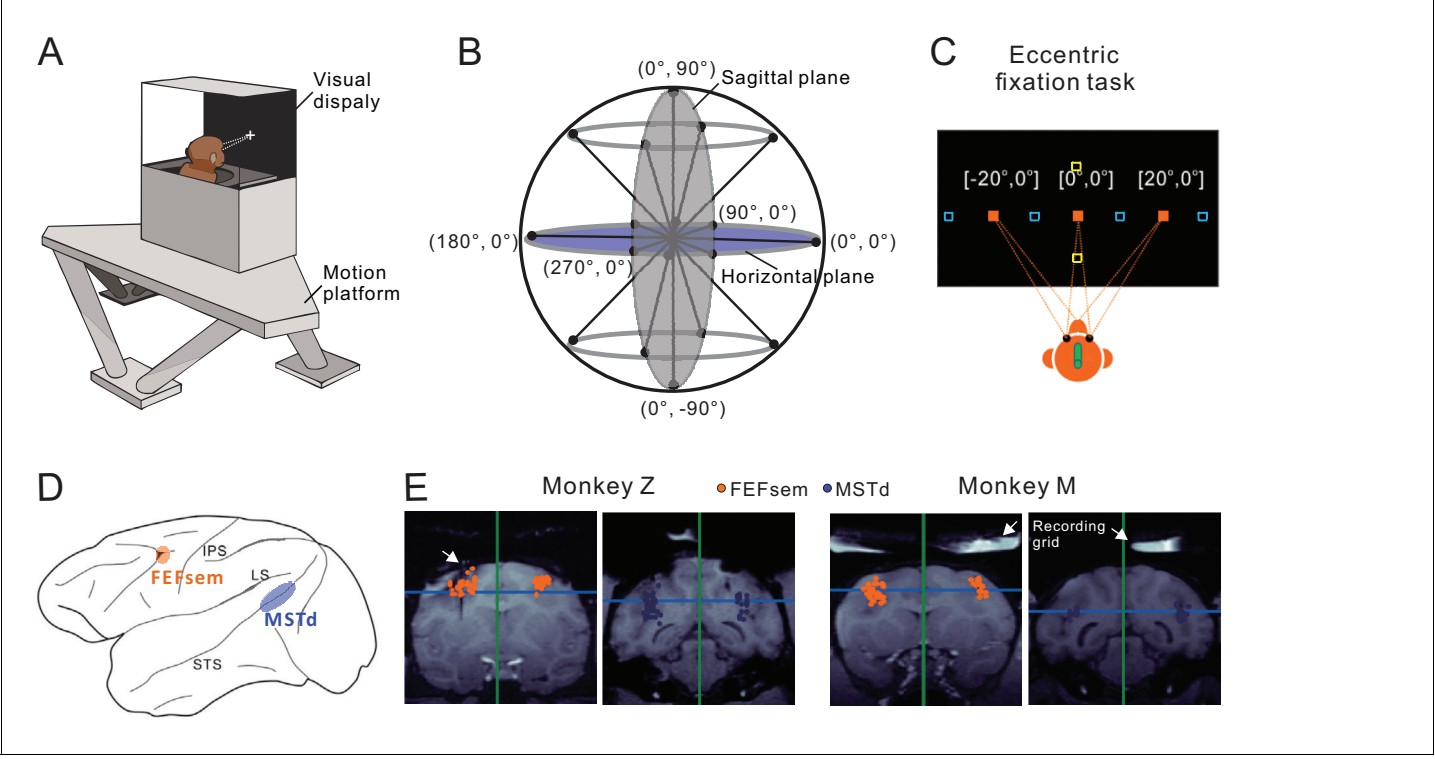

**Figure 1.** Experimental setup and anatomical locations. (**A**) Monkeys were trained to maintain fixation while seated in a virtual-reality setup. The apparatus consists of a 6-DOF motion platform that can translate in any direction. Visual display, monkey chair, and the field coil system are mounted on the motion base. Monkeys are head-fixed within the system. (**B**) Heading directions are varied in the horizontal and sagittal planes. (**C**) Eccentric fixation experimental paradigm. The fixation spot is presented at one of three locations: left (20°), center (0°), or right (20°) in the horizontal plane (filled origin square), or up (20°), center (0°), or down (20°) in the vertical plane (open yellow square). The open blue squares are fixation locations sometimes presented at ±30° or ±10° in an additional experiment. (**D**) Schematic illustration of the locations of the two cortical areas studied: FEFsem at the posterior portion of the arcuate sulcus, and MSTd at the posterior part of the superior temporal sulcus. (**E**) Coronal sections exhibiting the recording sites from the two monkeys (Monkey Z and M) in FEFsem (orange dots) and MSTd (blue dots). Note that all the recording sites are projected onto one single coronal plane, causing some points artificially appeared outside the region of interest (ROI). The white arrow in the first panel indicates the position of an electrode probe during MRI scanning. White arrows in the last two panels indicate the recording grid used to guide electrode penetrations in the current recording experiments.

DOI: https://doi.org/10.7554/eLife.29809.002

vestibular condition (*Figure 2A*, bottom panel). This pattern suggests that for this neuron, the visual signal is mainly eye-centered and the vestibular signal is mainly head-centered. To quantify this, we used two methods. The first method is to compute a displacement index (DI) based on the shift of the tuning curve that maximizes its correlation coefficient with the other tuning curves (see Materials and methods). A DI value of 1 means a complete shift of the tuning curve relative to the change in eye position (20°) and thus indicates an eye-centered reference frame. In contrast, a DI equal to 0 means zero-shift of tuning curves and indicates a head-centered reference frame. This example neuron has a DI value of 0.63 and 0.04 in the visual and vestibular condition, respectively, which is consistent with our intuition.

Across population in FEFsem (*Figure 2B*, *Figure 2—source data 1*), the mean visual DI is 0.74 ± 0.05 (mean ± s.e.m.), which is significantly different from either 0 (p=1.3E-32, t-test) or 1 (p=6.5E-7, t-test), suggesting that the visual signal is intermediate but more biased toward the eye-centered reference frame. In contrast, the mean vestibular DI is 0.14 ± 0.05, which is only slightly different from 0 (p=0.01, t-test), suggesting that the vestibular signal is predominantly head-centered. Between stimuli conditions, the mean visual DI is significantly larger than the mean vestibular DI (p=2.2E-7, t-test). Thus, overall the visual and vestibular reference frames in FEFsem are apart from each other. In MSTd (*Figure 2C*, *Figure 2—source data 2*), the vestibular signals are similar to those in FEFsem in that they are mainly head-centered: the mean vestibular DI is 0.17 ± 0.08 and is slightly

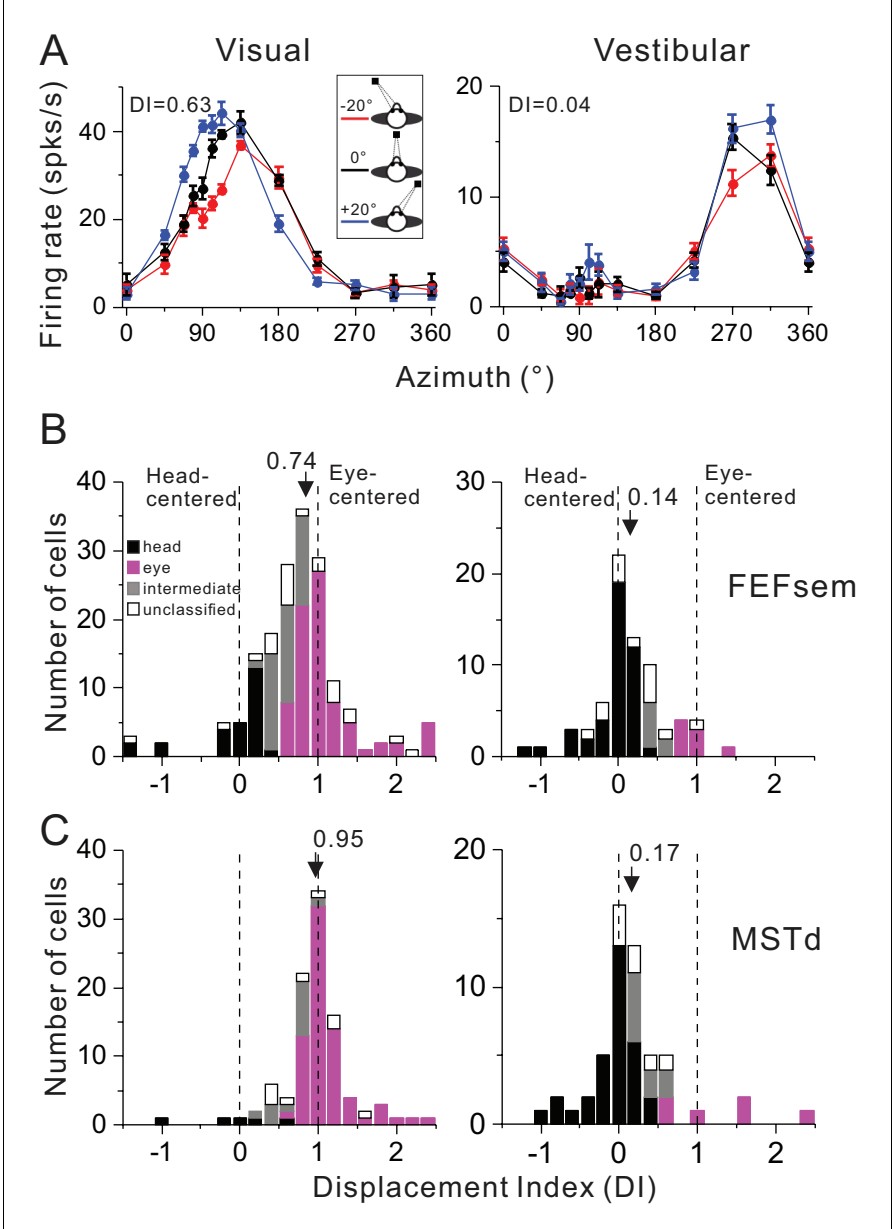

**Figure 2.** Summary of spatial reference frames as quantified by DI for visual and vestibular heading tuning measured in the eccentric fixation protocol. (**A**) Heading tuning functions of an example FEFsem neuron in the visual (left panel) and vestibular condition (right panel). Firing rate is plotted as a function of heading direction. Error bars are standard error of mean (s.e.m.). Different color curves represent tunings measured at different eye-in-orbit positions. (**B, C**) Distributions of DI measured under 20° eccentricity in FEFsem (**B**) and MSTd (**C**). DIs were limited in the range of [−1.5 2.5]. A few cases outside of this range were plotted at the edge of this range for the demo convenience. Black bars: head-centered coordinate; Magenta bars: eye-centered coordinate; Gray bars: intermediate coordinate; Open bars: unclassified coordinate. Arrowheads indicate the mean DI value of all the cases. Vertical dashed lines indicate head-centered (DI = 0) or eye-centered (DI = 1) coordinate.
DOI: https://doi.org/10.7554/eLife.29809.003

The following source data and figure supplements are available for figure 2:

**Source data 1.** Raw data for *Figure 2B*.
DOI: https://doi.org/10.7554/eLife.29809.007
**Source data 2.** Raw data for *Figure 2C*.
DOI: https://doi.org/10.7554/eLife.29809.008
**Figure supplement 1.** DI under different noise levels in hypothetical neurons.
*Figure 2 continued on next page*

*Figure 2 continued*

DOI: https://doi.org/10.7554/eLife.29809.004

**Figure supplement 2.** The relationship between DI and DDI in FEFsem (**A, C**) and MSTd (**B, D**).

DOI: https://doi.org/10.7554/eLife.29809.005

**Figure supplement 3.** DI measured under a broader range of eccentric fixation task by introducing two extra eccentricities of 10° and 30° in addition to the 20°.

DOI: https://doi.org/10.7554/eLife.29809.006

larger than 0 (p=0.03, t-test). As to the visual signals, the mean DI is 0.95 ± 0.05, which is not significantly different from 1 (p=0.3, t-test). This value is significantly larger than that in FEFsem (p=0.005, t-test), suggesting that the visual optic flow signals in MSTd are even closer to an eye-centered coordinate than in FEFsem. Hence, in general, the reference frames of visual and vestibular signals in both areas are largely separated, yet in FEFsem, they are slightly closer to each other (Δ mean DI$_{\text{vestibular, visual}}$ = 0.60) than in MSTd (Δ mean DI$_{\text{vestibular, visual}}$ = 0.78).

Since overall the DI values were broadly distributed, we computed the statistical 95% confidence interval (CI) of each cell through a bootstrap procedure (see Method). According to the CIs, each cell was categorized into one of the following four groups: (1) head-centered: CIs include 0 but not 1; (2) eye-centered: CIs include 1 but not 0; (3) intermediate: CIs are between 0 and 1 without touching 0 and 1; (4) unclassified: CIs belong to none of the above three types (e.g. including both 0 and 1). The last group usually reflects large noise in the tuning curves, but it only occupies a minor population in our data (*Figure 2B,C*). For visual signals in FEFsem, more cases are defined as the eye-centered group (46.8%) whereas some cases are defined as head-centered (15.8%) and intermediate (24.6%) group. In MSTd, there are even more cases that are defined as eye-centered group (71.7%) and fewer cases defined as the head-centered (5.1%) or intermediate (14.1%) group. As to the vestibular signals, in both areas, majority of the cases are defined as head-centered group (FEFsem: 60.6%; MSTd: 59.3%), and very few cases are defined as eye-centered (FEFsem: 11.3%; MSTd: 11.1%) and intermediate group (FEFsem: 9.9%; MSTd: 16.7%). To further explore how noise in the tuning curves may affect the DI measurement, we first ran simulations by creating populations of hypothetical neurons with different noise level (*Figure 2—figure supplement 1*). Our results show that large noise tends to broaden the DI distributions, but does not cause systematic bias toward a certain direction (e.g. head-centered). Indeed, in our real neuronal data, DI values are not significantly dependent on the noise level in the tuning curves (*Figure 2—figure supplement 2*). Thus, these results further support our above conclusions: the largely separated eye- and head-centered spatial coordinate respectively for the visual and vestibular signals are robust in FEFsem and MSTd.

We further measured DI under a broader range of eccentric fixation task by introducing two extra eccentricities of 10° and 30° in addition to the 20° used in the above experiment (see one example in *Figure 2—figure supplement 3A*). This allows us to examine whether spatial coordinate assessed by DI is dependent on the magnitude of the eccentric fixation amplitude, which has not been tested in previous works. Our result from a subpopulation of tested neurons (N = 38, data pooled across areas and stimuli conditions, *Figure 2—figure supplement 3B*) clearly shows that the average DI is not significantly different among eye positions with different amplitudes (p>0.3, t test), suggesting that using 20° of gaze eccentricity in our current study neither over- nor under-estimate the reference frames of cortical neurons.

The DI method conveniently gives intuition about the overall distribution of the spatial coordinate in each cortical area. To further examine how the tuning functions under different eccentric fixations for each individual cell can be best explained by the eye- versus head-centered models, we further employed a second method. Specifically, we simultaneously fit each neuron's tuning curves with modified wrapped Gaussian functions under all three eye positions with an eye-centered (prefer directions shifted as the varied eye positions) and a head-centered model (prefer directions unchanged) (see Materials and methods). *Figure 3A and B* show the model fitting results for the same example neuron as in *Figure 2*. Clearly the eye-centered model fits the data better in the visual condition (*Figure 3A*), while the head-centered model fits the vestibular data better (*Figure 3B*). To quantify this, the goodness-of-fit of each model measured by the partial correlation coefficient between the fit and the data was Z-scored to categorize each neuron's spatial coordinate (eye-centered versus head-centered, p<0.05, dotted lines in *Figure 3C,D*).

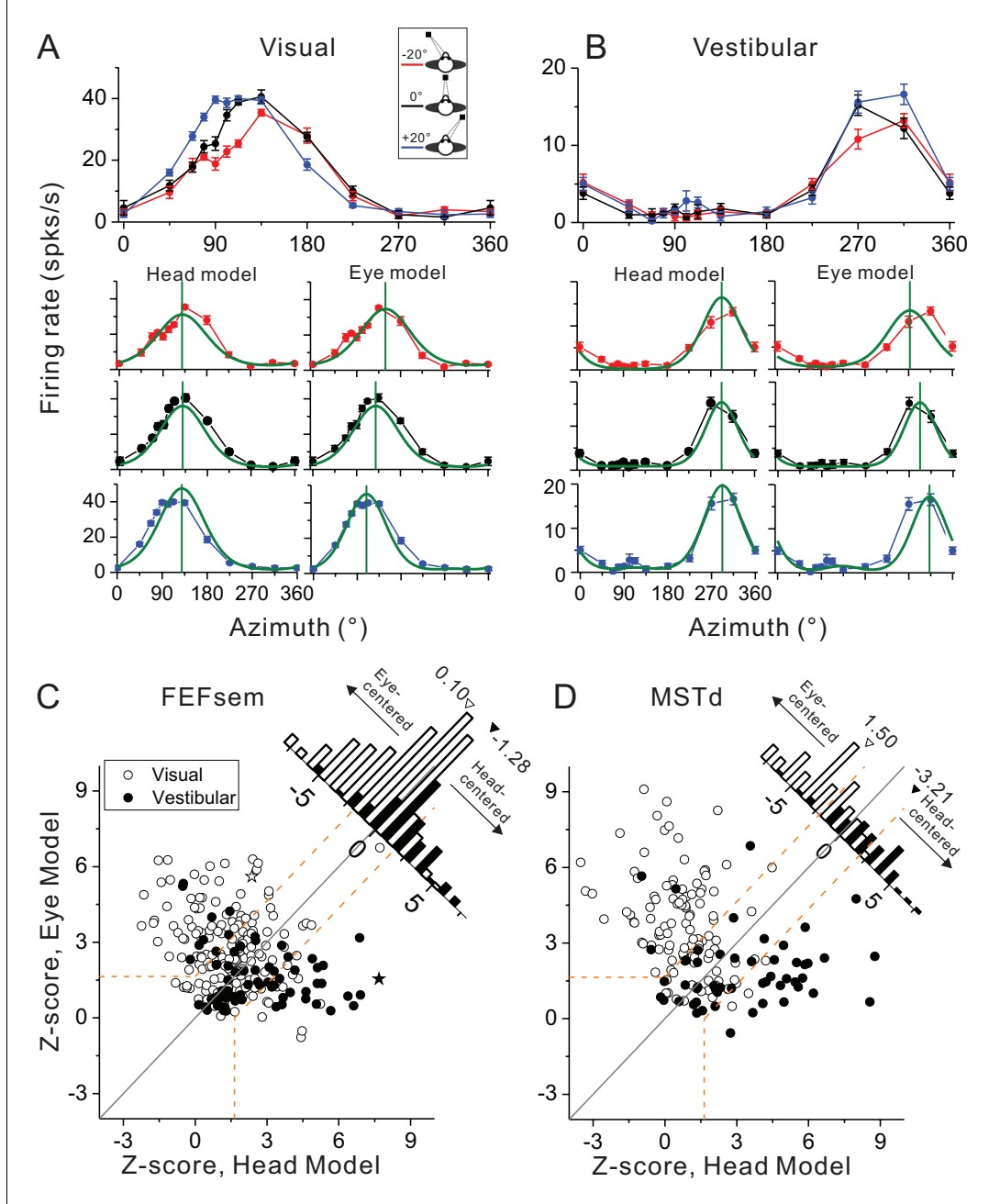

**Figure 3.** Spatial reference frames as assessed by head- and eye-centered model fittings. (A, B) Heading tuning functions from an example neuron in the visual (A) and vestibular (B) condition. Red, black and blue curves represent eccentric fixation at −20°, 0° and 20°, respectively. Superimposed green curves depict the fitting of eye-centered or head-centered models. Vertical green lines are the prefer directions of the fitting curves. (C, D) Eye- and head-centered model correlation coefficients (Z-transformed) in FEFsem (C) and MSTd (D). Filled symbols: vestibular; Open symbols: visual. Significance regions are based on the difference between eye- and head-centered Z scores corresponding to p<0.05 (top left, eye centered; bottom right, head centered; central diagonal region, intermediate). The two asterisks, one open and one filled, represent the example cell in (A) and (B), respectively. Diagonal histograms represent distributions of differences between Z-scores for each pair of models (eye- minus head-centered). Arrowheads indicate the mean values of each distribution. Filled bars: vestibular; Open bars: visual.

DOI: https://doi.org/10.7554/eLife.29809.009

Across population in FEFsem (*Figure 3C*), 36.3%, 7.0% and 56.7% neurons were classified as eye centered, head-centered and unclassified group in the visual condition, respectively. In the vestibular condition, 23.9%, 42.3%, 33.8% neurons were classified as eye-centered, head-centered and unclassified, respectively. The mean difference (eye - head) in Z-score between the two models is 0.10 and −1.28 in the visual and vestibular condition, respectively, which is significantly different from 0 in both cases (visual: p=0.0025, vestibular: p=2.0E-11, t-test). In MSTd, for the eye-centered, head-centered and unclassified category, there are 75.5%, 9.2% and 15.3% neurons respectively in the visual condition, and 24.1%, 61.1% and 14.8% neurons respectively in the vestibular conditions. The mean Z-score difference is 1.50 and −3.21 in the visual and vestibular condition respectively, both of which are significantly different from 0 (p=3.4E-4, p=1.4E-18,, t-test, *Figure 3D*). Hence, in line with the DI results, the model-fitting analysis also revealed that in both areas, generally there are more cases showing eye-centered reference frame for the visual signals, and more cases showing head-centered reference frame for the vestibular signals. However, compared between areas, the reference frames of the visual and vestibular signals are closer in FEFsem than in MSTd.

In the following, we further examined whether the spatial coordinates of the heading signals in FEFsem and MSTd were dependent on other factors including smooth pursuit eye movement, motion parallax cues in the visual optic flow, and the behavioral context for heading estimation.

## Smooth pursuit eye movement compensation

In addition to the static eye fixations varied at different eccentricities, other type of eye behavior, such as smooth pursuit is also frequently accompanied during spatial navigation which could potentially distort perceived flow field (*Warren and Hannon, 1988*, *1990*; *Royden et al., 1992*; *Banks et al., 1996*). In fact, some studies have used smooth pursuit paradigm to infer cortical neurons' spatial coordinate (*Bradley et al., 1996*; *Page and Duffy, 1999*; *Shenoy et al., 1999*, *2002*; *Ilg et al., 2004*; *Zhang et al., 2004*). However, recent studies argued that the more or less pursuit compensations as observed in these studies is not necessarily linked to spatial reference frames as measured under varied static eye positions during stimulus presentation (*Sunkara et al., 2015*). To test this, for some of the neurons that have been recorded under the eccentric fixation protocol, we further ran a pursuit block (*Figure 4A*, see Method). Briefly, the animals were required to pursue a moving target that was crossing the screen at a constant speed of 16°/s. During the steady pursuit, we presented visual optic flow stimuli that simulated real motion in the prefer plane as used in the eccentric fixation condition. Notice that in this protocol, the motion platform was always stationary such that there was no vestibular input to the animals.

*Figure 4B* shows two hypothetical (top panels) and two examples of real FEFsem (bottom panels) units tested under the pursuit protocol. Unlike the eccentric fixation protocol with unchanged gaze direction across the whole stimulus duration, continuous rotation of the eyes during pursuit causes shifts in tuning curves in a more complex way that is dependent on the heading preference of the neurons (*Sunkara et al., 2015*). Specifically, for neurons with lateral heading preference (0°/180°, top left panel in *Figure 4B*), tuning peak and trough remain unchanged. Instead, responses at the two sides of the tuning peak are shifted in opposite directions, causing the overall bandwidth increases or decreases for leftward or rightward pursuit, respectively. For neurons with forward/backward heading preference (90°/270°, top right panel in *Figure 4B*), the tuning peak and trough are shifted in opposite directions, also causing a change of the tuning width. Thus for each neuron, shift in the tuning curve was first computed separately from two parts: one between [0° 180°], and the other between [180° 360°], leading to a total of 4 values under the two pursuit directions. These values were then averaged to compute a single displacement index (DI) under the pursuit protocol, in a similar way as for the eccentric fixation task (see detail in Data Analysis). In this case, a DI value of 1 means complete tuning shift and implies that the neuron responds to resultant optic flow due to eye rotations. On the contrary, a DI value of 0 means unchanged tuning, and implies complete eye rotation compensation for representing true headings.

*Figure 4C* summarizes the results for 54 FEFsem neurons and 51 MSTd neurons (*Figure 4—source data 1*). The mean DI under pursuit is 0.24 ± 0.08 and 0.31 ± 0.10 in FEFsem and MSTd, respectively, both of which are substantially smaller than 1 (p=3.5E-13, p=3.0E-9, t-test) and slightly larger than 0 (p=0.002, p=0.002, t-test). In both areas, majority of the cells were classified as complete (FEFsem: 61.1%; MSTd: 39.2%) or intermediate compensation (FEFsem: 9.3%; MSTd: 35.3%). In contrast, there are very few cells exhibiting complete tuning shift (FEFsem: 5.6%; MSTd: 11.8%).

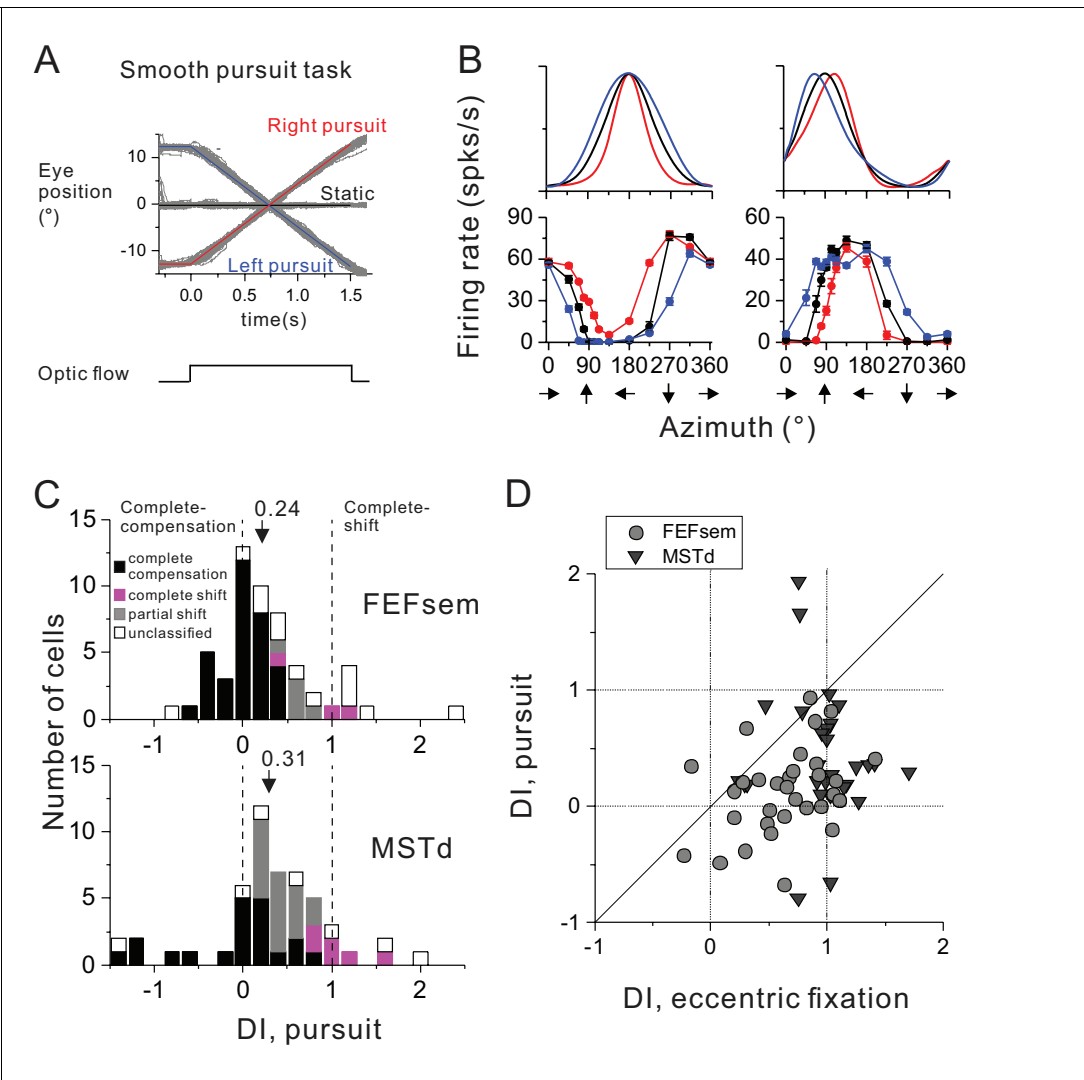

**Figure 4.** Comparison of spatial reference frames with smooth pursuit eye movement compensation. (**A**) Experimental paradigm for smooth pursuit eye movement. Monkeys were required to pursue a smooth moving target crossing the screen either from right to left (blue curve) or from left to right (red curve). These conditions were interleaved with a no-pursuit condition (central fixation, black curve). Gray curves are the raw eye traces from an example block. (**B**) Heading tuning curves from two hypothetical (top) and two example FEFsem (bottom) neurons under different pursuit conditions. Color is the same as in (**A**). (**C**) DI distributions under the pursuit protocol in FEFsem (upper panel) and MSTd (lower panel). Arrowheads indicate mean values. Vertical dashed lines indicate head-centered/complete compensation (DI = 0) or eye-centered/complete shift (DI = 1). Black bars: complete compensation for eye rotation; Magenta bars: complete shift from eye rotation; Gray bars: partial shift; Open bars: unclassified group. (**D**) Direct comparison of DIs between pursuit and eccentric fixation on a cell by cell basis. Each point represents one neuron. Circle: FEFsem, N = 36; Triangle: MSTd, N = 32.

DOI: https://doi.org/10.7554/eLife.29809.010

The following source data and figure supplement are available for figure 4:

**Source data 1.** Raw data for *Figure 4C*.
DOI: https://doi.org/10.7554/eLife.29809.012

**Figure supplement 1.** Direct comparison of DI between pursuit and eccentric fixation on a cell by cell basis.
DOI: https://doi.org/10.7554/eLife.29809.011

This result suggests that on average, both FEFsem and MSTd do not represent resultant optic flow under pursuit, but rather signal heading in a manner that is fairly tolerant to eye rotations.

Such a pattern is in sharp contrast to the spatial reference frame results under varied static eye positions (*Figure 2B,C*, left panel), suggesting that the two metrics are unlikely to be linked with each other. Indeed, a direct comparison on a cell by cell basis (*Figure 4D*) indicates that DI under

pursuit is significantly smaller compared to that under the eccentric fixation task (FEFsem: p=1.1E-4; MSTd: p=7.9E-5, paired t-test), and they are not significantly correlated with each other (FEFsem: p=0.10; MSTd: p=0.82 Spearman rank correlation). This conclusion holds even when we use identical method to compute DI values as for the eccentric fixation data (*Figure 4—figure supplement 1*). Thus, even complete eye rotation compensation for simulated heading observed in cortical neurons (e.g. FEFsem and MSTd) does not necessarily imply a head-centered spatial reference frame.

## Motion parallax cue in the visual optic flow

Depth cue such as motion parallax plays an important role in deciphering self-motion based on the depth structure of a visual scene (*Gibson, 1950*; *von Helmholtz, 1963*), yet this cue has not been included consistently across previous studies. In our above experiment, the visual optic flow has contained motion parallax cue. Thus in this section, we excluded this cue to test whether it might be a key to affecting the neurons' tuning shift under the eccentric fixation task (*Figure 5A–C*) or the smooth pursuit protocol (*Figure 5D–F*).

*Figure 5A* shows one typical neuron tested with and without motion parallax under the eccentric fixation protocol. Qualitatively the response pattern is similar under the two experimental conditions. This is also reflected in the population: the average DI is 0.80 ± 0.07 and 0.92 ± 0.06 in FEFsem and MSTd, respectively (*Figure 5B*, *Figure 5—source data 1*). In addition, majority of the cells are within the eye-centered category in FEFsem (60.5%) and MSTd (61.8%), whereas few cells are head-centered (FEFsem: 10.5%; MSTd: 2.9%) or intermediate (FEFsem: 15.8%; MSTd: 29.4%) reference frames. Such a result is very close to that under the condition with motion parallax cue (*Figure 2B, C*). When compared on a cell by cell basis (*Figure 5C*), DIs with and without motion parallax are highly correlated (R = 0.70, p=1.5E-7, Spearman rank correlation), and their means are not significantly different from each other (FEFsem: p=0.37; MSTd: p=0.29, paired t-test).

Under the pursuit protocol, excluding motion parallax slightly affect the neuronal responses (*Figure 5D–F*). The average DI is 0.26 ± 0.10 and 0.45 ± 0.14 in FEFsem and MSTd, respectively (*Figure 5E*, *Figure 5—source data 2*). The proportion of complete compensation, complete shift, and intermediate cells is 48.7%, 7.7% and 33.3%, respectively, in FEFsem, and 31.6%, 23.7% and 23.7%, respectively, in MSTd. Again, these results are similar to those under the condition with motion parallax cues (*Figure 4C*). Compared on a cell by cell basis (*Figure 5F*), DIs with and without motion parallax are significantly correlated (FEFsem: R = 0.68, p=2.9E-4, MSTd: R = 0.74, p=4.4E-6, Spearman rank correlation), and the mean DI under the no-motion parallax cue condition is slightly but significantly larger than that under the motion parallax contion (FEFsem: p=0.01; MSTd: p=0.07, paired t-test). Thus, similar to the effect in the eccentric fixation protocol, excluding motion parallax from the visual stimuli has limited effect on FEFsem and MSTd's tolerance of the eye rotations.

## Behavioral context for heading estimation

In previous studies, spatial reference frames have been measured under the condition in which the animals passively experienced heading stimuli during eccentric fixations (*Avillac et al., 2005*; *Fetsch et al., 2007*; *Takahashi et al., 2007*; *Chen et al., 2013*; *Fan et al., 2015*). Thus, it is possible that the predominant eye-centered reference frame of the visual optic flow may have been overestimated, because under this behavioral context, the brain may not use these signals for heading judgment based on a head coordinate. To test this hypothesis, we introduced an active behavioral paradigm in the current study (see Materials and methods). Briefly, the animals were trained to perform a heading estimation task in which they were required to report perceived headings simulated from optic flow by making oculomotor responses from center to a peripheral ring that appeared at the end of each trial (*Figure 6A*). The tricky part in this task was that the headings were varied in the whole horizontal plane, while the ring was presented on the frontal parallel plane. As a result, the animals needed to correctly associate the headings with their oculomotor responses: left/right headings correspond to left/right saccade, and importantly, forward/backward headings correspond to upward/downward saccade. In each trial, the center of the presented ring was always aligned with the fixation location during heading presentation.

*Figure 6B,C* show one monkey's performance under three eccentric fixation conditions (−20°, 0° and 20°) in one experimental session. Compared to the heading stimuli, the animal showed more or less estimation error in a manner qualitatively similar to human being's performance (*Crane, 2015*,

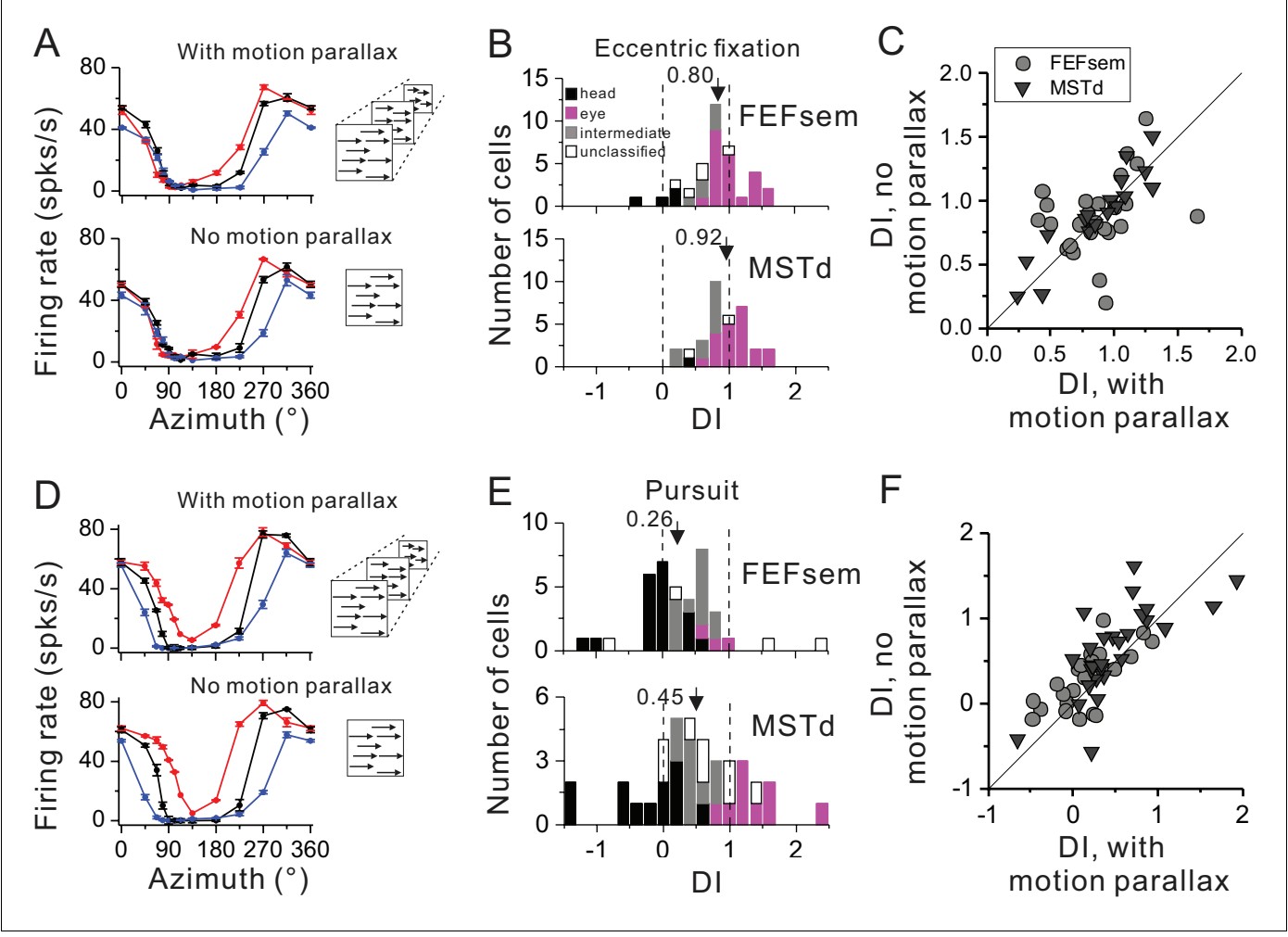

**Figure 5.** Influence of motion parallax on spatial reference frame (A–C) and pursuit compensation (D–F) measurements. (A) Visual tuning functions of an example FEFsem neuron with and without motion parallax in the optic flow. Red, black and blue curves represent eccentric fixation at −20°, 0° and 20°, respectively. (B) Distribution of DI without motion parallax cue in FEFsem (upper panel, N = 38) and MSTd (bottom panel, N = 34). Black bars: head-centered coordinate; Magenta bars: eye-centered coordinate; Gray bars: intermediate coordinate; Open bars: unclassified coordinate. Arrowheads indicate the mean value. (C) Direct comparison of DIs between with and without motion parallax cues on a cell by cell basis. Each symbol represents a neuron. Circle: FEFsem, N = 27; Triangle: MSTd, N = 17. (D–F) Same format as in (A–C) but for the pursuit protocol. Red, black and blue curves represent rightward, fixation only (no-pursuit) and leftward pursuit, respectively. In (E), for FEFsem, N = 39; for MSTd, N = 38; In (F), for FEFsem, N = 24; for MSTd, N = 29.
DOI: https://doi.org/10.7554/eLife.29809.013

The following source data is available for figure 5:

**Source data 1.** Raw data for *Figure 5B*.
DOI: https://doi.org/10.7554/eLife.29809.014
**Source data 2.** Raw data for *Figure 5E*.
DOI: https://doi.org/10.7554/eLife.29809.015

*2017*). Notice though, across the whole training process, the estimation errors were large (~20°) at the early phase (*Figure 6D*), suggesting the animals judged headings simulated from optic flow relative to their eye positions. After trained with feedback signals of reward for a week or two, this error was reduced to a few degrees, indicating that the animals learned to judge headings largely based on a head-centered coordinate (*Figure 6D*). We then started collecting neural data from the two monkeys after their behavioral performance reached a plateau. Specifically, across the whole recording period, the average heading estimation error was only less than ~20% (3.6 ± 0.52° and

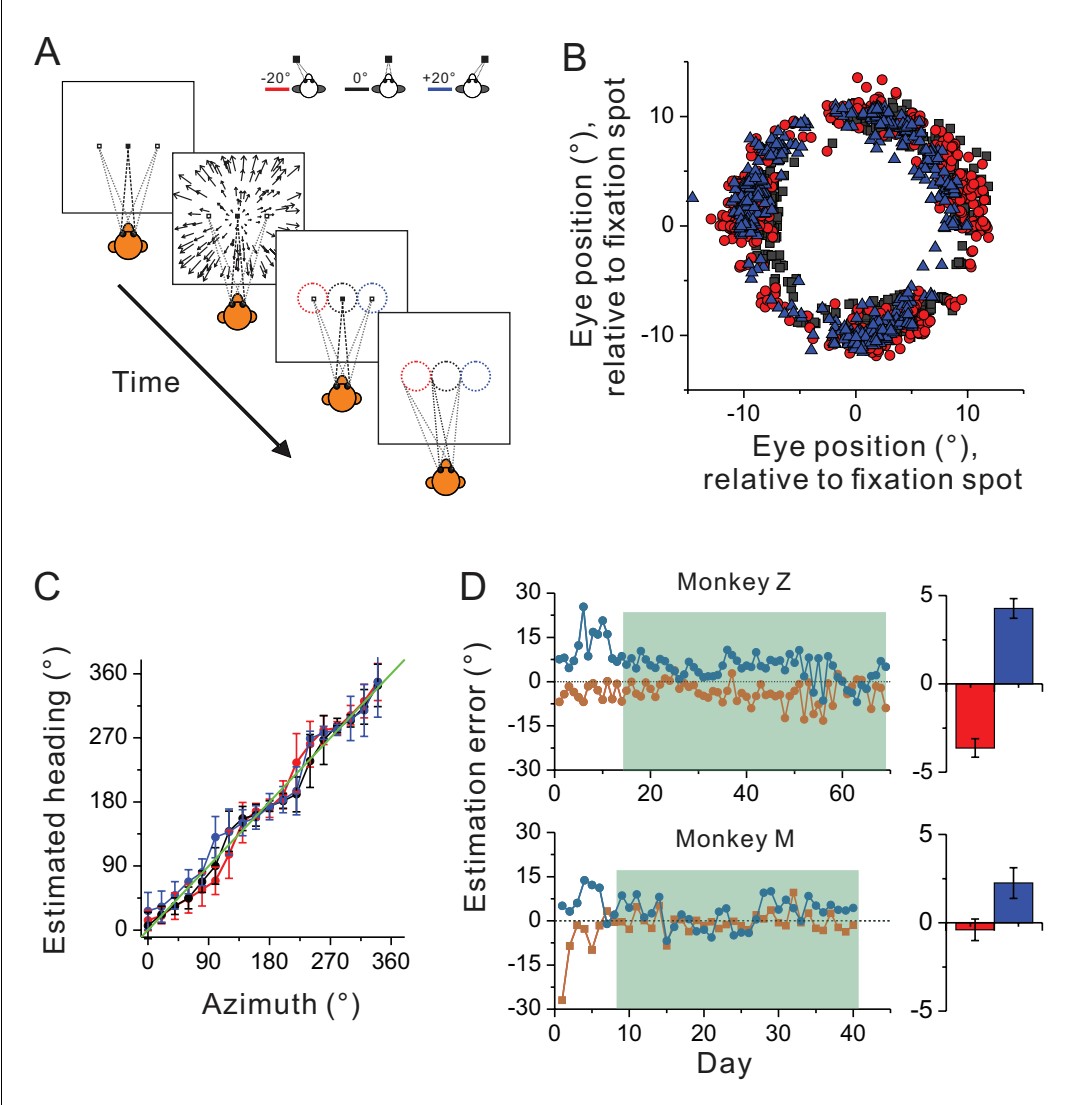

**Figure 6.** Heading estimation task in the visual condition. (**A**) Schematic illustration of the heading estimation task. The four panels depict the event sequence in one trial. Each trial begins with one of the three fixation locations (−20°, 0°, or 20°). After capturing fixation, visual optic flow is provided simulating heading in the horizontal plane. Headings are varied with a resolution of 20°, spanning a full 360° range. At the end of the trial, a target ring appears for ocular motor response. The ring is made of 36 dots apart by 10°. It is 20° in diameter, and its center is aligned with the fixation location in each trial (−20°, 0°, or 20°). Saccade endpoints within a window of 5 × 5° are taken as correct choice and the animals will be rewarded. Red, black and blue symbols represent eccentric fixation at −20°, 0° and 20°, respectively, and are the same for the rest of the figure. (**B**) Behavioral data from one experimental session with 30 repetitions for each stimulus condition. Note that saccade endpoints are plotted relative to the center of the target ring instead of the visual screen, thus data from the three fixation locations are roughly overlapped in this plot. (**C**) Mean and circular SD of the monkey's heading estimates are plotted as a function of the real heading direction. The green diagonal line represents perfect performance. (**D**) Performance of the two monkeys in the training (unfilled areas) and recording sessions (shaded areas). Heading estimation errors are evaluated by computing the difference in the heading estimate between the eccentric and central fixation conditions. Right histograms represent the mean ± s.e.m. of the data in the recording sessions (shaded area).

DOI: https://doi.org/10.7554/eLife.29809.016

4.3 ± 0.56°) in monkey Z, and less than ~10% (0.4 ± 0.62° and 2.2 ± 0.87°) in monkey M (marginal bar graphs corresponding to the shaded area in *Figure 6D*).

For each neuron, we collected data under two blocks: passive viewing condition (i.e. fixation only) and active estimation condition (i.e. oculormotor response). We first compared the visual spatial coordinates under the fixation only condition before and after the animals were trained with the estimation task (*Figure 7A*, *Figure 7—source data 1*). Interestingly, we found after training, there was a

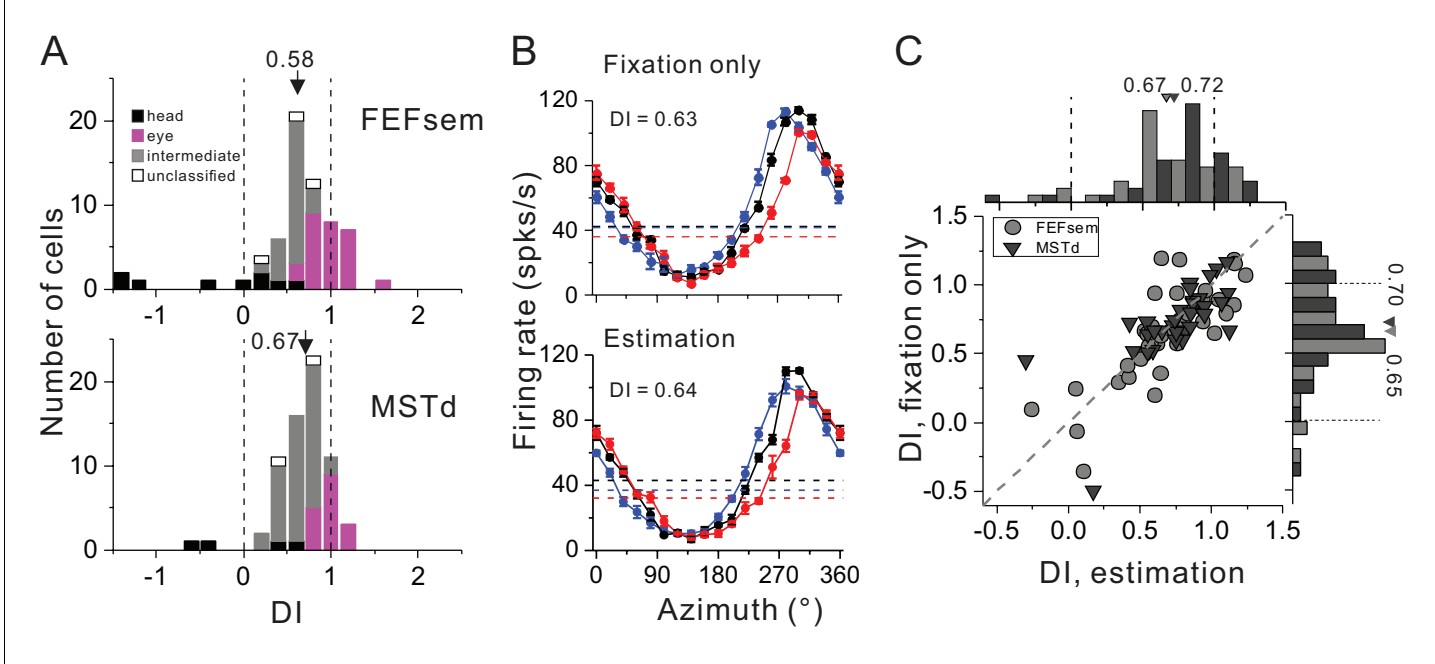

**Figure 7.** Spatial reference frame of cortical neurons during heading estimation task. (**A**) DI distributions post training of heading estimation task in FEFsem (upper panel: N = 65) and MSTd (bottom panel: N = 68). Black bars: head-centered coordinate; Magenta bars: eye-centered coordinate; Gray bars: intermediate coordinate; Open bars: unclassified coordinate. Arrowheads indicate mean values. Vertical dashed lines indicate head- (DI = 0) or eye-centered (DI = 1) coordinate. (**B**) Tuning functions of an example FEFsem neuron during the fixation only (post training) and active estimation conditions. (**C**) Comparison of DIs between fixation only and active estimation conditions on a cell by cell basis. Circle: FEFsem, N = 36; Triangle: MSTd, N = 35.

DOI: https://doi.org/10.7554/eLife.29809.017

The following source data is available for figure 7:

**Source data 1.** Raw data for *Figure 7A*.
DOI: https://doi.org/10.7554/eLife.29809.018

tendency that the mean DI became smaller in both FEFsem (after: 0.58 ± 0.07, before: 0.74 ± 0.05, p=0.08, t-test) and MSTd (after: 0.67 ± 0.04, before: 0.95 ± 0.05, p=1.6E-5, t-test). Compared between the two areas, the training effect seems to be more obvious in MSTd: the population of the eye-centered cells was reduced by about 40% (after: 26.5%, before: 71.7%) and the intermediate population was increased by about 50% (after: 66.2%, before: 14.1%). By contrast in FEFsem, after training, the eye-centered population remained almost the same (after: 41.5%, before: 46.8%) while the intermediate population was increased by about 15% (after: 40.0%, before: 24.6%). Notice though, the degree of this coordinate shift in MSTd resulted from training is much limited such that the visual signals are still largely separated from the head-centered coordinate.

We next assessed and compared whether after training, the visual reference frame would be different under the passive viewing and the active estimation conditions (*Figure 7B,C*). For example, *Figure 7B* shows one example neuron exhibiting similar response patterns under the two behavioral conditions (passive viewing: DI = 0.63; active estimation: DI = 0.64). This pattern also holds across population on a cell by cell basis (*Figure 7C*): DIs are highly correlated (FEFsem: R = 0.81, p=2.3E-9; MSTd: R = 0.77, p=7.3E-8, Spearman rank correlation) and their means are not significantly different from each other (FEFsem: p=0.59; MSTd: p=0.55, paired t-test). Hence, after training, the spatial coordinates of the visual optic flow in both MSTd and FEFsem are not significantly affected by the behavioral context.

## Gain modulation

So far our results have suggested that the multisensory heading signals may not share a common reference frame. Then how does the brain combine the visual and vestibular signals to represent

heading that is based on a head reference during spatial navigation? One possibility is that the brain may employ a 'gain field' mechanism for coordinate transformations as proposed in previous computational studies (*Zipser and Andersen, 1988*; *Xing and Andersen, 2000*). For example, MSTd units' activities under optic flow are modulated by different eye positions, and these eye position signals, together with the eye-centered visual responses, could be combined by downstream neurons to implement coordinate transformation, leading to a head-centered heading representation (*Gu et al., 2006*). In this section, we further examine the gain field property in FEFsem and compare it with that in MSTd.

To quantify the gain field, we computed the difference in the maximum evoked response at different eye positions. The maximum evoked response in each tuning curve was the maximum mean firing rate subtracted by the minimum activity. Thus, any difference in this metric across eccentric fixation conditions would mainly reflect a gain modulation effect rather than an additive eye position effect. *Figure 8A–D* show four example neurons with significant gain modulations. The first two neurons' activities are monotonically increased or decreased as a function of eye positions (*Figure 8A, B*), and are thus defined as 'monotonic' group. In contrast, the second two neurons show increased (*Figure 8C*) or decreased (*Figure 8D*) activities at central fixation compared to the responses at eccentric eye positions. These neurons are defined as 'non-monotonic' group.

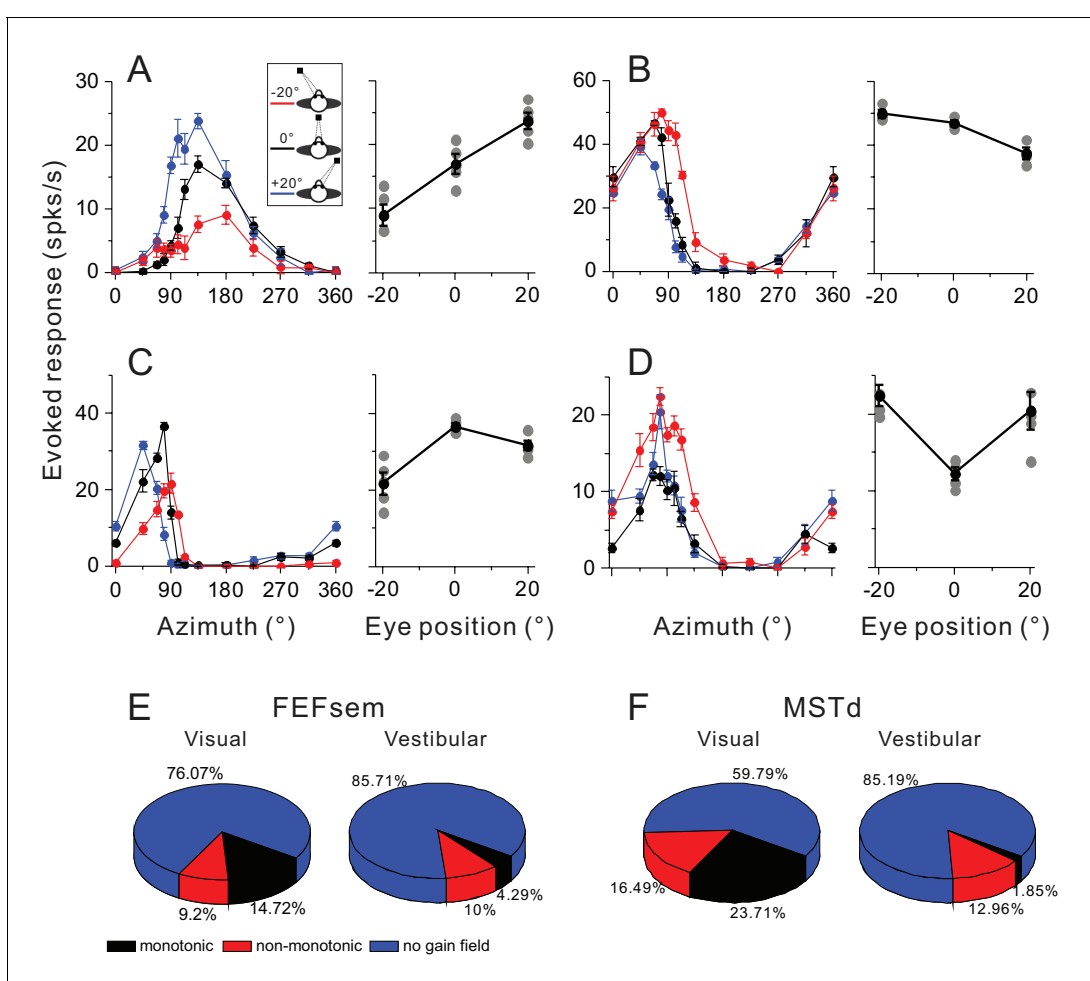

**Figure 8.** Gain modulations in FEFsem and MSTd. (A–D) Four example units with significant gain modulations across fixation locations. A and B show two cases with monotonic gain effect. In C and D, the effect is not monotonic, but rather that the responses are either strongest or weakest during central fixation. Tuning curves are the evoked responses with the minimum mean firing rate subtracted. Red, black and blue symbols represent eccentric fixation at −20°, 0° and 20°, respectively, and are the same for the rest of the figure. (E, F) Proportion of different category of neurons in the visual and vestibular conditions in FEFsem (E) and MSTd (F). Red: monotonic group; Black: non-monotonic group; Blue: no-gain modulations.
DOI: https://doi.org/10.7554/eLife.29809.019

In FEFsem, overall there are 23.9% neurons showing significant gain modulation in the visual condition (*Figure 8E*). Among them, 9.2% and 14.7% neurons belong to the 'non-monotonic' and 'monotonic' category, respectively. This pattern is similar to that in MSTd: 40.2% neurons show significant gain modulation, with 16.5% and 23.7% neurons in the 'non-monotonic' and 'monotonic' category, respectively. Notice that these results are acquired under only three eccentric eye positions varied in one axis. Thus the proportion of cells with significant gain modulation, especially for the monotonic category, might have been underestimated. Indeed, in a previous study when the eye positions were varied in multiple points in a two dimensional plane, more than 80% neurons in MST were found to be modulated by eye positions, and among them, majorities exhibited a 'monotonic' effect (*Bremmer et al., 1997b1997b*).

In any case, less than half of the neurons in both FEFsem and MSTd contain gain fields that could potentially be used by downstream neurons to transform the eye-centered visual signals into a head- or medial coordinate. Similarly, in the vestibular condition, there are also a number of neurons with gain fields, although its proportion is relatively smaller compared to the visual signals (FEFsem: 14.3%; MSTd: 14.8%). Hence, downstream neurons could also potentially use these signals to transform the vestibular signals into a mediate coordinate that better match the reference of visual signals.

## Discussion

In the current study, we measured spatial coordinate of visual and vestibular heading signals in two cortical areas of FEFsem and MSTd. Compared to MSTd which is mainly a mid-stage sensory area, FEFsem is at a later stage along the dorsal visual pathway and is closer to the motor side (*Felleman and Van Essen, 1991*). Consistent with this notion, we discover that the visual reference frame in FEFsem is significantly skewed towards the head centered coordinate by roughly 20% compared to that in MSTd. However, the average visual DI in FEFsem (0.74) is still largely separated from the vestibular DI (0.14), suggesting the spatial coordinates of the two heading signals in FEFsem are distinct. This result is robust as the measured visual coordinate is independent on a number of factors including smooth pursuit eye movements and motion parallax cue in the optic flow. Interestingly, training the animals to judge heading directions relative to their head slightly shifts the reference frame of the visual signals towards the head centered coordinate in both areas. However, after training, the reference frames of the two heading signals remain fairly separated under both passive and active behavioral contexts. Hence, neither MSTd nor FEFsem has fully completed coordinate transformations of visual and vestibular signals for multisensory heading perception. Other sensory regions need to be explored in future experiments, but with proper methods including complete tuning function measures under active behavioral performance.

### Factors confounding spatial coordinate measures

The visual optic flow is generated due to self-moving in the environment, and the focus of expanding flow (FOE) has a zero velocity that can be used to estimate the translation direction of the body, that is, heading (*Tanaka et al., 1986*; *Warren and Hannon, 1988*; *Duffy and Wurtz, 1995*; *Britten, 2008*). However, this information is based on the retina, and it is often confounded by different eye behavior during spatial navigation. For example, fixating at an eccentric target during forward moving will shift the FOE on the retina and subsequently may cause bias in heading estimate when solely relying on the visual information. In laboratory, researchers have designed experimental paradigms by varying eye positions while measuring heading performance in human subjects (*Crane, 2015*, *2017*), or neuronal tuning functions in animals (*Fetsch et al., 2007*; *Chen et al., 2013*; *Sunkara et al., 2015*). On the other hand, some researchers have adopted smooth pursuit eye movement that is also frequently accompanied during spatial navigation and could distort the optical flow field in either human psychophysical studies (*Warren and Hannon, 1988*; *1990*; *Royden et al., 1992*; *Banks et al., 1996*) or neurophysiological studies on monkeys (*Bradley et al., 1996*; *Page and Duffy, 1999*; *Shenoy et al., 1999*; *Ilg et al., 2004*; *Zhang et al., 2004*). In general, these works reported that both behavior and neurons could compensate for smooth pursuit eye movements and recover true heading simulated from optic flow, implying a head-centered spatial reference frame.

Although the above two types of eye behavior share similar properties such as varied gaze, they dramatically differ in several aspects. First, on the behavioral level, the eccentric fixation protocol has a fixed eccentric gaze across the stimulus duration, and its impact on the shift of FOE projected on the retina is solely determined by and equal to the magnitude of the gaze eccentricity. By contrast in the pursuit condition, the gaze direction is continuously changed across the stimulus duration, and its impact on the resultant optic flow is not related with the momentary eye positions, but rather a number of other factors including pursuit speed, flow speed and flow depth (*Zhang et al., 2004*). Second, on the neuronal level, tuning curves are simply shifted consistently in one direction if they represent a predominant eye-centered spatial coordinate in the eccentric fixation protocol. Instead in the pursuit condition, tuning curves in the whole plane (instead of in a limited heading range) are expected to shift in a more complex way if they represent the resultant optic flow on the retina.

In our current work, for the first time to our knowledge, we have measured tuning curves from the same population of neurons in both eccentric fixation and smooth pursuit protocols. We discover that there is no significant relationship between the two metrics computed from the two experimental conditions. Our results demonstrate that full or near full eye rotation compensation does not have to imply a head centered spatial reference frame. In another word, the pursuit compensations are not necessarily linked to reference frames as measured under the varied static eye positions, which is consistent with the opinion proposed in a recent study (*Sunkara et al., 2015*).

In addition to the type of different eye behavior, the depth cue of the motion parallax has also been used inconsistently in previous studies. In our current work, we have measured the tuning curves in both with and without motion parallax conditions. Our results indicate that motion parallax does not affect the reference frame measures. However, there is a weak yet statistically significant effect in the smooth pursuit compensation. This effect is sort of expected since the induced FOE shift under pursuit is also determined by the depth of the optic flow (*Zhang et al., 2004*). In our experiment, the motion parallax cue in the optic flow simulates a cube of dots (40 cm in depth) symmetrically crossing the fixation plane. Thus, the pursuit effect on heading perception may roughly be similar, but not identical between the 2-dimensional flow restricted in one single plane (fixation plane) and the 3-dimensional flow across multiple planes. On the other hand, motion parallax has been suggested to play an important role in deciphering rotational (due to pursuit) and translational components of self-motion in the optic flow field. Thus it is somehow surprising that we have not observed too much difference in the neural activity of cortical neurons under the motion parallax conditions. It is possible that a much larger impact of motion parallax could be observed in a simulated pursuit condition that has been missing in our current experimental design (*Bremmer et al., 2010*). Future experiments including both real and simulated pursuit protocols need to be conducted to examine this hypothesis.

## Behavioral contexts

A recent study shows that human subjects estimate heading directions irrelevant of the eye positions under the vestibular condition, whereas in the visual condition, the eccentric gaze causes roughly 46% shift in the perceived heading (*Crane, 2015*). This result suggests that heading perception based on visual optic flow may be biased somehow towards retina coordinate. However, there is no feedback signals provided to the subjects in this study, thus it is unclear how feedback signals and learning process may help recover true headings under eccentric fixations. Indeed, we found that the monkey's perceived heading was largely affected by eye positions initially. After training the animals to judge headings relative to the head by rewarding for a week or two, the estimation bias was reduced to only a few degrees, roughly 10 –20% of the gaze magnitude, implying a predominant head coordinate.

On the neuronal level, we assess the training effect mainly from three aspects. First, after training, the visual reference frame in FEFsem and MSTd is slightly shifted towards the head coordinate. However, this change is modest (~20%). Second, once trained, the visual reference frame does not show significant difference under the passive viewing and active estimation tasks. Third, compared to the passive viewing task, the overall response magnitude is increased or decreased on a minor proportion of neurons (11.3% and 8.5%, respectively) under the active estimation task. Thus generally training/learning seems to have a limited effect on the visual coordinate measured in the current two brain areas (FEFsem and MSTd). However, it remains possibility that this effect may be larger in

other brain regions such as the ventral intraparietal area (VIP, [*Chen et al., 2011b*]) and the visual posterior sylvian fissure (VPS, [*Chen et al., 2011a*]). In future experiments, proper methods need to be employed including complete tuning curve measures under active heading estimation contexts.

### Distinct visual and vestibular spatial coordinate in sensory cortices

The distinct visual and vestibular spatial coordinate is potentially an obstacle for cue integration. This is because if the heading information conveyed from each sensory cue is confounded by different types of eye behavior that are frequently accompanied during natural navigation, it would be hard to imagine how the brain could integrate inconsistent sensory evidence across different modalities in a statistically optimal way (*Deneve and Pouget, 2004*). One straightforward intuition is that somewhere in the brain, the visual and the vestibular heading signals are transformed into a common reference frame. Computational works have proposed that this is feasible through integration of the eye position signals: the neural network could either transform one coordinate to the other (e.g. from eye to head, or from head to eye), or transform both coordinates to an intermediate one (*Zipser and Andersen, 1988*; *Siegel, 1998*; *Gu et al., 2006*; *Fetsch et al., 2007*). In these cases, the hidden units in the model exhibit a 'gain' field as observed in many cortical areas including the premotor area (*Pesaran et al., 2006*), parietal area of 7a (*Zipser and Andersen, 1988*; *Siegel, 1998*; *Xing and Andersen, 2000*), V6A (*Hadjidimitrakis et al., 2014*), V6 (*Fan et al., 2015*), VIP (*Chen et al., 2013*) and MSTd (*Gu et al., 2006*; *Fetsch et al., 2007*). In our current work, we have also observed that roughly a quarter of FEFsem neurons exhibit gain fields for the visual optic flow signals. Interestingly, this number is relatively smaller compared to that in MSTd (~40%). Considering the fact that the overall visual coordinate of FEFsem is more intermediate than in MSTd (*Figure 2B, C*), these results may suggest that compared to extrastriate visual cortex, FEFsem is at a later stage for spatial coordinate transformations.

For downstream areas that receive both eye-centered visual optic flow and eye position signals for coordinate transformations, we expect to observe a head- or near head-centered coordinate of the heading signals in these regions. However, no such areas have been discovered so far. For all the sensory areas that researcher have explored including VIP (*Chen et al., 2013*), MSTd (*Gu et al., 2006*; *Fetsch et al., 2007*), V6 (*Fan et al., 2015*) and even FEFsem in the current study, majority of neurons exhibit fairly eye-centered visual optic flow signals, and most of them carry eye position signals at the same time. Thus, overall these areas may still serve as an intermediate stage for coordinate transformations. In the future, one strategy is to keep searching for head-centered visual reference frames in the other sensory cortices. On the other hand, coordinate transformations may never be accomplished in the sensory cortices, instead they may be implemented in the sensory-motor association areas in which decisions and ocular motor responses are formed for multisensory heading perception. So the other strategy is probably to study the posterior parietal cortex, prefrontal cortex, or the subcortical area of the superior colliculus.

## Materials and methods

### Animal preparation

Surgical preparation and training have been described in detail in previous studies (*Gu et al., 2006*; *Fetsch et al., 2007*; *Takahashi et al., 2007*; *Chen et al., 2013*). Briefly, two male rhesus monkeys (Macaca mulata), weighing 6–10 kg were chronically implanted, under sterile conditions, with a lightweight plastic head-restraint ring that was anchored to the skull using titanium inverted T-bolts and dental acrylic. The ring was 5–6 cm in diameter, serving as the head post and recording chamber at the same time. Scleral search coil was implanted in one eye for tracking and measuring eye movements in a magnetic field. After surgical recovery, behavioral training was accomplished using standard operant conditioning procedures through water/juice reward. Before recording experiments, a plastic grid containing staggered rows of holes (0.8 mm spacing) was stereotaxically secured inside the head ring covering majority of the space, allowing for accessing multiple areas at the same time. The grid was positioned in the horizontal plane. Vertical microelectrode penetrations were made via transdural guide tubes inserted in the grid holes. All procedures were approved by the Animal Care Committee of Shanghai Institutes for Biological Sciences, Chinese Academy of Sciences (Shanghai, China).

## Anatomical localization

FEFsem was initially identified via a combination of structural MRI scans and the pattern of eye movements evoked by electrical microstimulation (~50 μA, 200 Hz) as described previously (*Gu et al., 2016*). Briefly, the arcuate sulcus in the frontal lobe was first identified to locate FEF. Electrical stimulation was then applied to evoke eye movements for distinguishing the subregion of FEFsem from FEFsac (*MacAvoy et al., 1991*; *Gottlieb et al., 1994*). Specifically, microstimulation in FEFsem usually evoked smooth eye movements to the ipsilateral direction with the recording hemisphere, which was in contrast to the fast saccade eye movement in the direction opposite to the recording hemisphere. Within FEFsem, we further used a pursuit protocol to verify whether the isolated single unit was indeed a pursuit neuron. In particular, the animals were required to pursue a fixation spot that moved linearly in one of 8 equally-spaced directions in the frontoparallel plane at a speed of 20°/s (*Gottlieb et al., 1994*; *Gu et al., 2016*). Only neurons significantly tuned under the pursuit protocol (p<0.05, One-way ANOVA) were further recorded for other parts of experiments in the current study.

Extracellular single-unit recordings were performed with tungsten microelectrodes (tip diameter 3 μm, impedance 1–2 MΩ at 1 kHz, FHC, Inc.) that was advanced into the cortex through a transdural guide tube, using a micromanipulator (FHC, Inc.). Single neurons were isolated using a conventional amplifier and a dual voltage-time window discriminator (Bak Electronics, Mount Airy, MD). The times of occurrence of action potentials and all behavioral events were recorded with 1 ms resolution by the data acquisition computer. Raw neural signals were also digitized at 25 kHz and stored to disk for off-line spike sorting (CED Spike2, UK). To allow a direct comparison of the response properties between FEFsem and MSTd on the same animals, we also recorded neurons in MSTd. MSTd was identified using procedures similar to those in previous studies (*Gu et al., 2006*; *Fetsch et al., 2007*). Briefly, MSTd was at the posterior tip of the superior temporal sulcus (AP: ~−2 mm, ML:~15 mm). MSTd neurons usually had large receptive fields that contained the fovea and part of the ipsilateral visual field. MSTd neurons were also sensitive to visual motion defined by global optic flow stimuli. After reaching MSTd, if advancing electrode further down (in the vertical way) for another few millimeters, the middle temporal area MT was usually encountered with neurons containing much small receptive fields that were typically in the contralateral visual field.

## Behavioral task and experimental procedures

Translation of the monkeys in 3D space was accomplished by a motion platform (MOOG 6DOF2000E; Moog, East Aurora, NY). During experiments, the monkey was seated comfortably in a primate chair, which was secured to the platform and inside the magnetic field coil frame. A LCD-screen was mounted on the motion platform (subtending 90 × 90° of visual angle) placed 30 cm in front of the monkey (*Figure 1A*). The screen and the field coil frame were mounted on the motion platform. In order to activate vestibular otolith organs, each transient inertial motion stimulus followed a smooth trajectory with a Gaussian velocity profile, providing the 'vestibular' stimulus condition. In the 'visual' condition, global optic flow that occupied the whole screen was provided to simulate self-motion through a 3D random dot field (OpenGL graphics library). All the dots were moving coherently (100%), generating a strong motion signal.

Heading stimuli including eight directions equally apart were first delivered in two planes: horizontal and sagittal planes (*Figure 1B*). A 'preferred' plane was then chosen for each isolated neuron with significant (p<0.05, One-way ANOVA) and strongest modulation. Five experimental blocks were subsequently applied: (1) eccentric fixation in visual (with motion parallax), and/or vestibular condition, (2) eccentric fixation in visual condition without motion parallax, (3) smooth pursuit in visual condition with motion parallax, (4) smooth pursuit in visual condition without motion parallax, and (5) active heading estimation task. In general, in all experiments, the animals were required to maintain fixation at the fixation spot within an electronic window (2 × 2°). Exceeding the window would result in abandon of the trial.

### Eccentric fixation task

The monkeys were presented with a fixation spot (0.2° in diameter) at one of three locations: one central, that is straight forward (0°), and two peripheral targets with equal eccentricity of 20° either in the horizontal or vertical axis (*Figure 1C*). In some additional experiments, seven possible fixation

locations were introduced and interleaved including:±30,±20,±10, and 0°. The monkey initiated each trial by acquiring fixation within a 2 × 2° electronic window. After fixation, visual and/or vestibular heading stimuli were presented in the preferred plane that was determined in advance. Heading typically included eight directions that were spaced at 45° intervals, leading to a full range of 360°. In the horizontal plane, four additional directions apart by 11.25° were interpolated around straight ahead (0 ± 11.25°, 0 ± 22.5°). Each stimulus condition was repeated at least three times, yet majority of the neurons (92%) were collected for five or more repetitions. Visual and vestibular heading stimuli had same Gaussian velocity profile (duration: 2 s; travelling distance: 0.11 m; peak acceleration: 0.85 m/s$^2$; peak velocity: 0.25 m/s). The visual stimulus had two conditions, either with motion parallax (namely '3D'), or without motion parallax (namely '2D'). In the '3D' case, the virtual workspace was 100 cm wide, 100 cm high, and 40 cm deep. Star density was 0.01/cm$^3$, with each star being a 0.15 × 0.15 cm triangle. Thus the visual stimulus simulates the animal's approaching a 3D cloud of dots. In the '2D' case, the flow dots were distributed only within the fixation plane with a density of 0.4/cm$^2$, whereas all the other experimental parameters were identical as in the '3D' case. Stimuli were viewed binocularly but without disparity information (no stereo cues). The animals were required to maintain fixation during the stimulus duration of 2 s while passively experiencing heading stimuli. At the end of the trial, the animals were rewarded with a drop of liquid after successful fixation.

### Smooth pursuit

During the visual optic flow presentation, the animals were required to pursue the fixation spot that was moving smoothly at a constant speed of 16°/s. The pursuit direction could be either in the horizontal or the vertical axis. Pursuit started from an eccentric position of 12 °, and the average eye position during a pursuit trial was aligned with the center of the screen. The two pursuit directions and a control condition of central fixation were randomly interleaved in one experimental block. Similar to the eccentric fixation task, the monkeys were required to maintain gaze within a 2 × 2° eye window during pursuit, except that at the first and last 250 ms of a trial, this window is larger (4 × 4°). Spikes in these two windows were excluded from the analysis window (see data analysis). Monkeys were rewarded after accomplishing pursuit at the end the trial.

### Heading estimation task

In this context, the animals were required to actively report their perceived headings simulated from visual optic flow by making saccade to a choice ring that appeared at the end of the trial (*Figure 6A*). The simulated heading directions were varied at 20° intervals in the horizontal plane. The animals maintained fixation at one of the three locations (−20°, 0°, 20°) across the stimulus duration. At the end of the trial, the fixation spot disappeared, and the choice ring, 20° in diameter, appeared at the screen. The center of the ring was always aligned with the fixation location in a certain trial, which could be at one of the three fixation locations (−20°, 0°, 20°). The ring consisted of 36 dots uniformly spaced at 10° intervals. Notice that the headings and the choice targets were in different planes, thus, the animals needed to correctly associate their perceived headings (in the horizontal plane) to the choice targets (in the fronto-parallel plane). Specifically, lateral headings corresponded to lateral choice targets, and forward/backward headings corresponded to upward/downward targets. The size of the reward window was 5 × 5°. Each block consisted of 54 conditions (18 headings × 3 eye positions). Each condition was repeated at least 5 times, leading to more than 270 trials. In addition, another control block was included in which the animals only maintained fixation during stimulus presentation and were not required to make choice at the end of trial. Thus, the total trials in this experiment was >540 (270 × 2).

## Data analysis

All data analyses and statistical tests were performed using MATLAB (MathWorks, Natick, MA). Tuning profiles for the different stimulus and task conditions were constructed by plotting the mean firing rate (spikes/s) as a function of heading direction. Firing rate was computed over the middle 1 s of each successfully completed trial. This analysis window was chosen because of two reasons. First, most of the velocity variation occurred in the central 1 s of Gaussian velocity profile (*Gu et al.,*

*2006*). Second, this is the time period when the animal's pursuit eye movement is stable in the smooth pursuit protocol.

## Displacement index (DI) under eccentric fixation task

For each pair of tuning curves (p<0.05, One-way ANOVA) at two different eye positions, the amount of shift is quantified by computing the cross-covariance metric (*Avillac et al., 2005*; *Fetsch et al., 2007*):

$$DI_{ij} = \frac{k^{max\left(cov\left[R_i(\theta), R_j(\theta+k)\right]\right)}}{P_i - P_j} \tag{1}$$

, where $k$ (in degrees) is the relative displacement of the tuning functions (denoted $R_i$ and $R_j$), and the superscript above $k$ refers to the maximum covariance between the tuning curves as a function of $k$ (ranging from $-180°$ to $+180°$). Tuning functions were linearly interpolated with a resolution of 1°. The denominator represents the difference between the eye positions ($P_i$ and $P_j$) under which the tuning functions were measured. If the shift between a pair of tuning curves is equal to the change in eye position, the DI equals to 1, meaning eye-centered coordinate. If tuning curves are not shifted at all, DI will be 0, meaning head-centered coordinate. Any single neuron is included in the dataset if it has at least two tuning curves (at two of the three fixation locations) that passed the statistical criterion (p<0.05, One-way ANOVA). If all tuning curves were significant, there would be three DI values (combinations from three fixation locations), and a single averaged DI is assigned for the unit. For each DI value, the 95% confidence intervals were computed from a bootstrap resampling procedure. Briefly, bootstrapped tuning functions under each eccentric fixation condition were obtained by resampling (with replacement) the same number of repetition from the original tuning functions. A new DI value was computed from the new tuning curves across the eccentric fixation conditions. This process was repeated 1000 times, producing a corresponding distribution of DIs from which 95% confidence intervals could be derived.

## Displacement index (DI) under smooth pursuit protocol

DI was also used to quantify tuning shift under the smooth pursuit protocol. However, in this case, the simple cross-correlation method as used in the eccentric fixation task was not sufficient to characterize all of the changes in the tuning curves (*Sunkara et al., 2015*). Instead, a 3-step partial shift analysis procedure was applied (*Sunkara et al., 2015*). (1) Peak-to-trough modulations in the tuning curves of pursuit trials were linearly scaled to match that in the no-pursuit trials). (2) Each tuning function was split into two halves: one with heading ranges [0° 180°], and one [180° 360°]. The split tuning curve needed to be significant (p<0.05, One-way ANOVA), and was linearly interpolated to a resolution of 1°. (3) Within each half heading range, tuning curve under the no-pursuit condition was circularly shifted (in step of 1°) to maximize the correlation coefficient between the two comparison tuning curves. Thus, each neuron has four shift values at most (two pursuit conditions plus two heading ranges). These values are averaged to serve as the numerator in Equation [1]. The denominator is the predicted shift of tuning curves that respond to resultant optic flow under eye rotations, which is roughly 30° in our case (*Bradley et al., 1996*; *Shenoy et al., 2002*; *Zhang et al., 2004*).

## Eye- and head-centered models

Tuning curves were fit with a modified wrapped Gaussian function (*Fetsch et al., 2007*) of the following form:

$$R(\theta) = A_1 \cdot \left[ e^{\frac{-2 \times \left(1 - cos\left(\theta - \theta_p\right)\right)}{\sigma^2 \cdot \kappa}} + A_2 \cdot e^{\frac{-2 \times \left(1 - cos\left(\theta - \theta_p - 180°\right)\right)}{\sigma^2}} \right] + R_0 \tag{2}$$

There are six free parameters: $\theta_p$ is the peak location, $\sigma$ is the tuning width, $A_1$ is the overall amplitude, and $R_0$ is the baseline response level. The second exponential term produces a second peak 180° out of phase with the first peak, but only when $A_2$ is sufficiently large. This extra term is necessary for fitting a small population of neurons with more than one peak. The relative widths of the two peaks are determined by the parameter $\kappa$. The goodness of fit is quantified by the

correlation coefficients of $R^2$ between the fitting and the raw data. To eliminate bad fits from our analysis, we excluded a minority of data (~4%) with $R^2 < 0.6$.

Tuning curves under the three eccentric fixation task ($-20°$, $0°$, $20°$) are fit with an eye-centered and a head-centered model simultaneously (using the Matlab function fmincon). Thus, the total number of free parameters in each fitting is 16 (3 eye positions × 5 free parameters of $A_1$, $\sigma$, $\kappa$, $A_2$ and $R_0$ in Equation [2] + $\theta_0$). $\theta_0$ is the eccentric fixation task in the eye-centered model ($\theta_0$ for $0°$ fixation, $\theta_0 - 20°$ for leftward fixation, and $\theta_0 + 20°$ for rightward fixation). In the head-centered model, $\theta_0$ is always the same across eye positions.

For each fit, the correlation coefficient between the fit and the data is used to assess the goodness-of-fit. To remove the influence of correlations between the models themselves, partial correlation coefficients are calculated with the following formula (*Fetsch et al., 2007*):

$$R_e = \frac{(r_e - r_h r_{eh})}{\sqrt{(1 - r_h^2)(1 - r_{eh}^2)}}$$
$$R_h = \frac{(r_h - r_e r_{eh})}{\sqrt{(1 - r_e^2)(1 - r_{eh}^2)}}$$

(3)

, where $r_e$ and $r_h$ are the correlation coefficients between the data and the eye- and head-centered models, respectively. $r_{eh}$ is the correlation between the two models. Partial correlation coefficients $R_e$ and $R_h$ are normalized using Fisher's r-to-Z transform so that Z-scores from two the models are independent of the number of data points (*Angelaki et al., 2004*; *Smith et al., 2005*; *Fetsch et al., 2007*). A criterion of 1.645 for the Z-score (equivalent to p=0.05) is used to categorize cells into three groups: eye-centered (*Figure 3C,D*, dashed lines).

## Acknowledgement

We thank Wenyao Chen for monkey care and training, Ying Liu for software programming. This work was supported by grants from the National Natural Science Foundation of China Project (grant 31471048), the National Key R and D Program of China (2016YFC1306801), and the Strategic Priority Research Program of CAS (XDB02010000).

## Additional information

### Funding

| Funder | Grant reference number | Author |
|---|---|---|
| National Natural Science Foundation of China | 31471048 | Yong Gu |
| National Key R&D Program of China | 2016YFC1306801 | Yong Gu |
| Chinese Academy of Sciences | Strategic Priority Research Program (XDB02010000) | Yong Gu |

The funders had no role in study design, data collection and interpretation, or the decision to submit the work for publication.

### Author contributions

Lihua Yang, Resources, Data curation, Software, Formal analysis, Investigation, Visualization, Methodology, Writing—original draft; Yong Gu, Supervision, Funding acquisition, Investigation, Visualization, Project administration, Writing—review and editing

### Author ORCIDs

Yong Gu http://orcid.org/0000-0003-4437-8956

### Ethics

Animal experimentation: All animals in this research are lawfully acquired and their retention and use are in every case in compliance with national and local laws and regulations, and in accordance with

the Guide for Care and Use of Laboratory Animals of Instituted for Laboratory Animal Research. All experimental procedures are approved by the Animal Care Committee of Shanghai Institutes for Biological Science, Chinese Academy of Science (ER-SIBS-221409P for Professor Yong Gu).

## Decision letter and Author response

Decision letter https://doi.org/10.7554/eLife.29809.021
Author response https://doi.org/10.7554/eLife.29809.022

## Additional files

### Supplementary files

• Transparent reporting form
DOI: https://doi.org/10.7554/eLife.29809.020

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
