## [Decision Letter]

Thank you for submitting your article "Distinct spatial coordinate of visual and vestibular heading signals in macaque FEFsem and MSTd" for consideration by *eLife*. Your article has been reviewed by three peer reviewers, and the evaluation has been overseen by a Reviewing Editor and a Senior Editor (Richard Ivry).

The reviewers have discussed the reviews with one another and the Reviewing Editor has drafted this decision to help you prepare a revised submission.

Summary:

The paper studies the representation of visual and vestibular heading in primate cortical areas MSTd and FEFsem. Besides providing novel data for FEFsem, an area for which a heading representation has only recently been described, the main point of the paper is a comparison of references frames (eye-centered vs head-centered) in the two areas and for the two modalities. Results show that the representation is mostly modality specific (eye-centered for optic flow and head-centered for vestibular signals) in both areas. All three reviewers were very positive about the interest of the question, the quality of the experimental approach and the overall presentation of the data.

Essential revisions:

The conclusions of the manuscript strongly depend on the DI statistics. The DI seems suitable for separating between head and eye reference frames in the sense that responses in head reference frames will tend to have smaller values than responses in the eye reference frame. Therefore, it supports the main conclusion of the manuscript. However, the reviewers are concerned that estimation of the DI might be noisy and the exact values of the DI might be biased depending on how well the tuning curves are estimated. Biases in the estimation of the DI might be especially problematic when comparing between conditions. For example, could the difference in DI between FEF and MSTd (Figure 2) result from the difference in the response size of the cell and not a true difference in the reference frames? Could the difference between pursuit and fixation (Figure 4) be a result of the difference in the magnitudes of the responses – or the difference in the way the DI are computed? One way to partly overcome this would be to calculate, when possible, the DI from the population average rather than for each cell separately. Population averages would also provide further important information about the magnitude and pattern of the responses. Alternatively, the authors could perform bootstrapping with their data to assess the statistical significance of DI, or perform simulations to support the use of DI.

The statistical analysis in subsection “Behavioral context for heading estimation” is incorrect. The authors use a one tailed paired t-test; however, a paired t-test may be used only when comparing values that were recorded in the same neurons. Here the authors compare neuronal responses recorded two weeks apart. These can't originate from the same neurons since they author don't perform chronic recordings; therefore the authors can't use a paired t-test. Furthermore, there is no justification for using a one tailed test in this context. The authors should use the correct statistics (two-tailed unpaired test) and update their results accordingly.

Regarding the investigation of smooth pursuit and motion parallax: the most interesting condition is missing. Theoretically, motion parallax is needed to distinguish rotational and translational components of self-motion in the optic flow field. Rotational components are introduced by smooth pursuit. To test the ability of the visual system to deal with rotational components based on flow analysis, researchers have used the paradigm of simulated pursuit, in which the observer fixates while the flow simulates both a translational heading and a rotational pursuit (see for example, Bremmer et al., 2010). It is conceivable that a much larger impact of motion parallax could be observed in a simulated pursuit condition. The reviewers realize that this condition was not part of the experiment, and do NOT think it is necessary to provide this data. However, the limitations of the study in that regard should be discussed. It may explain why excluding motion parallax from the visual stimuli has limited effect on FEFsem and MSTd's tolerance of the eye rotations (subsection “Motion parallax cue in the visual optic flow”).

[Editors' note: further revisions were requested prior to acceptance, as described below.]

Thank you for resubmitting your work entitled "Distinct spatial coordinate of visual and vestibular heading signals in macaque FEFsem and MSTd" for further consideration at *eLife*. Your revised article has been favorably evaluated a Reviewing editor along with three reviewers, and overseen by Richard Ivry as Senior Editor.

The reviewers have evaluated the revised version and found that all their comments have been addressed except for one point. The reviewing editor has helped draft the following summary of how we would like to see this issue addressed.

The concern is that the DIs computed in the eccentric fixation and smooth pursuit tasks might be biased towards 0 for cells with noisy tuning curves. Therefore, cells with a lower response magnitude would have a lower signal to noise ratio and the DI would be biased towards zero. As a result, differences between the visual and vestibular cases could result from differences in the magnitude of the responses and not from differences in the coordinates. The bootstrapping methods performed by the authors would not eliminate this bias since the bootstraps would also contain the bias. To test if the DI statistics are biased, the authors could perform simulations in which they construct noiseless tunings curves with a DI of one and then add different levels of noise, calculating the DI in the exact same way they calculate the DI for neurons.

If it turns out that DI is biased, it would still be possible to show that this bias does not affect conclusions. This would entail controlling for the magnitude of the response. For example, calculate the DI as a function of the tuning magnitude. This would allow you to examine the following:

1) Cells with the same magnitude of response have a DI close to one during visual condition and close to 0 in vestibular condition.

2) Cells with the same magnitude of the response during fixation and pursuit have a different DI.

The authors could also use Figure 3 as a control.

To sum up, the authors should, as a first step, run simulations to check if the DIs are biased towards 0 when the signal/noise ratio is low. If this is the case, they should demonstrate that it does not affect their conclusions (for instance using the approach suggested above).

---

## [Author Response]

The paper studies the representation of visual and vestibular heading in primate cortical areas MSTd and FEFsem. Besides providing novel data for FEFsem, an area for which a heading representation has only recently been described, the main point of the paper is a comparison of references frames (eye-centered vs head-centered) in the two areas and for the two modalities. Results show that the representation is mostly modality specific (eye-centered for optic flow and head-centered for vestibular signals) in both areas. All three reviewers were very positive about the interest of the question, the quality of the experimental approach and the overall presentation of the data.

We thank reviewers for their thoughtful comments on our work which have spurred us to perform additional data analyses and revise the context to make the paper much stronger. We have attempted to respond to each comment with new analyses or text revisions, as detailed below. The revised file is provided in two versions: one clean file, and one with highlighted changes. We feel that these major additions, combined with many other revisions to address specific points of the reviewers, make the paper much stronger and we hope that the reviewers will now find it acceptable for publication.

Essential revisions:The conclusions of the manuscript strongly depend on the DI statistics. The DI seems suitable for separating between head and eye reference frames in the sense that responses in head reference frames will tend to have smaller values than responses in the eye reference frame. Therefore, it supports the main conclusion of the manuscript. However, the reviewers are concerned that estimation of the DI might be noisy and the exact values of the DI might be biased depending on how well the tuning curves are estimated. Biases in the estimation of the DI might be especially problematic when comparing between conditions. For example, could the difference in DI between FEF and MSTd (Figure 2) result from the difference in the response size of the cell and not a true difference in the reference frames? Could the difference between pursuit and fixation (Figure 4) be a result of the difference in the magnitudes of the responses – or the difference in the way the DI are computed? One way to partly overcome this would be to calculate, when possible, the DI from the population average rather than for each cell separately. Population averages would also provide further important information about the magnitude and pattern of the responses. Alternatively, the authors could perform bootstrapping with their data to assess the statistical significance of DI, or perform simulations to support the use of DI.

We understand the reviewers’ concern and appreciate their constructive suggestions. To address this question, we performed a bootstrapping procedure to compute the 95% confidence interval for each DI value. We then labeled each cell’s categorization, i.e. eye-, head-, intermediate or unclassified according to this measurement in all the main figures of DI distributions. In particular, the last group indicates cells with noisy tuning curves. Fortunately, this group only occupies a minor proportion. We can clearly see that our previous main results and conclusions remain consistent. In another word, our original observations of the difference in the DI across experimental conditions are not due to the different noise or magnitude in each condition. We have now added this information in the Material and methods and Results sections. With this additional analysis, we are hoping that it can release the reviewers’ concerns.

As to the reviewer’s concern about the different way of computing DI between the pursuit and fixation conditions (Figure 4), we re-calculated DI values for the pursuit data with the identical method as in the fixation condition (see Figure 4—figure supplement 1). Specifically, DI for pursuit was not computed from each half of the tuning curve, but rather from the whole tuning curve at once as for the eccentric fixation data. Obviously, the results and conclusions remain similar (the new Figure 4—figure supplement 1).

The statistical analysis in subsection “Behavioral context for heading estimation” is incorrect. The authors use a one tailed paired t-test; however, a paired t-test may be used only when comparing values that were recorded in the same neurons. Here the authors compare neuronal responses recorded two weeks apart. These can't originate from the same neurons since they author don't perform chronic recordings; therefore the authors can't use a paired t-test. Furthermore, there is no justification for using a one tailed test in this context. The authors should use the correct statistics (two-tailed unpaired test) and update their results accordingly.

Thank you for capturing this mistake. We actually have used independent t test but have mistakenly described as “paired t test” in the old text. We now corrected this error. We also agree with the review in that it is not justified to use one-tailed test here, so we have changed it to the two-tailed test. Not surprisingly, this increased the p value a bit, which was 0.08, and we thus modified our wording in the text accordingly (subsection “Behavioral context for heading estimation”).

Regarding the investigation of smooth pursuit and motion parallax: the most interesting condition is missing. Theoretically, motion parallax is needed to distinguish rotational and translational components of self-motion in the optic flow field. Rotational components are introduced by smooth pursuit. To test the ability of the visual system to deal with rotational components based on flow analysis, researchers have used the paradigm of simulated pursuit, in which the observer fixates while the flow simulates both a translational heading and a rotational pursuit (see for example, Bremmer et al., 2010). It is conceivable that a much larger impact of motion parallax could be observed in a simulated pursuit condition. The reviewers realize that this condition was not part of the experiment, and do NOT think it is necessary to provide this data. However, the limitations of the study in that regard should be discussed. It may explain why excluding motion parallax from the visual stimuli has limited effect on FEFsem and MSTd's tolerance of the eye rotations (subsection “Motion parallax cue in the visual optic flow”).

We totally agree with the reviewer and appreciate this comment. We realized about this limit in our methodology, and have included it in the Discussion section.

[Editors' note: further revisions were requested prior to acceptance, as described below.]

[…] To sum up, the authors should, as a first step, run simulations to check if the DIs are biased towards 0 when the signal/noise ratio is low. If this is the case, they should demonstrate that it does not affect their conclusions (for instance using the approach suggested above).

As the reviewer suggested, we first ran simulations as illustrated in the new Figure 2—figure supplement 1. Our results show that as the signal/noise ratio is low in the tuning curves, the distribution of DI across hypothetical neurons becomes broader, generating some cases of seemingly head-centered coordinate (DI=0). However, there are three points indicating that this pattern is different from that for the real neurons: (1) the number of the case toward 0 is small compared to the majority of the cases with DI=1 (eye-centered); (2) the distribution of DI is symmetric around 1, i.e. extending in both directions instead of systematically biased toward 0; (3) the seemingly head-centered cases under the low signal/noise ratio situations turned out to be “unclassified” group instead of “head-centered group” based on the bootstrap test (open bars, top marginal distributions in Figure 2—figure supplement 2).

Indeed, as shown in the real neurons (new Figure 2—figure supplement 2), there is no significant relationship between the signal/noise ratio and DIs. In fact, many cases carrying similar signal/noise ratio exhibit clearly separated eye-centered (DI=1) and head-centered (DI=0) coordinate for the visual and vestibular signals, respectively. Similarly, the signal/noise ratio is analogous under the pursuit and eccentric fixation condition (the pursuit signal/noise ratio is even slightly higher, which is in the opposite direction as the reviewer’s concern), the pursuit DDI is still close to 0 and the eccentric fixation DDI is close to 1.

Based on these analyses, we believe that the noise in the tuning curves cannot change the main conclusions in the current paper. In another word, the vestibular signals are mainly head centered and the visual signals are more eye-centered in cortices. Under pursuit, the visual optic flow signals are more tolerant to eye rotations. We have incorporated these new results in the text (subsection “Spatial coordinate of visual and vestibular signals”).